# The Relationship Between Neurocognitive Function and Concussion in Women Professional Football Players: A Cross-Sectional Study

**DOI:** 10.3390/sports13120448

**Published:** 2025-12-11

**Authors:** Freja Fredrika Lähteenmäki, Steve den Hollander, Dina Christa Janse van Rensburg, Tuomas Brinck, Gino Kerkhoffs, Vincent Gouttebarge

**Affiliations:** 1Department of Orthopedic Surgery and Sports Medicine, Amsterdam University Medical Center Location, University of Amsterdam, Meibergdreef 9, 1105AZ Amsterdam, The Netherlands; f.f.lahteenmaki@amsterdamumc.nl (F.F.L.);; 2Amsterdam Collaboration on Health & Safety in Sports (ACHSS), IOC Research Center, 1105AZ Amsterdam, The Netherlands; 3Football Players Worldwide (FIFPRO), 2132LR Hoofddorp, The Netherlands; 4Division of Physiological Sciences and Health Through Physical Activity, Lifestyle and Sport Research Centre, Department of Human Biology, Faculty of Health Sciences, University of Cape Town, Rondebosch, Cape Town 7701, South Africa; 5Section Sports Medicine, Faculty of Health Sciences, University of Pretoria, Pretoria 0028, South Africa; christa.jansevanrensburg@up.ac.za; 6Department of Orthopaedics and Traumatology, Mehiläinen Hospitals, 00100 Helsinki, Finland; 7Academic Center for Evidence-Based Sports Medicine (ACES), 1105AZ Amsterdam, The Netherlands; 8Amsterdam Movement Sciences, Ageing & Vitaly, Muscuvloskeletal Health, Sports, 1105AZ Amsterdam, The Netherlands

**Keywords:** football, concussion, neurocognition, women, sport

## Abstract

Objective: To determine the neurocognitive functions of women professional football players and explore their potential connection to concussions. Methods: An observational cross-sectional study was conducted via electronic questionnaires. Neurocognitive function was assessed with the “CNS Vital Signs” testing tool. Results: In total, 68 participants performed the neurocognitive function testing. Compared with the reference population, players scored within the average range (≥90) for 11 of 12 neurocognitive domains. Motor speed was above average (SS = 111.7). Overall, no significant neurocognitive deficits were observed. Thirty-two participants (43%) reported one or more concussions, with defenders being most affected (50%). Among defenders, 64% (*n* = 16) have a history of one or multiple concussions. Players with a history of three concussions showed significant deficits in the simple attention domain. Conclusion: Professional women footballers did not show significant signs of neurocognitive function deficits. However, a history of three concussions was significantly associated with lower standard scores for the simple attention neurocognitive domain.

## 1. Introduction

Neurocognition is defined as cognitive functions involving different domains, such as memory, attention and executive functioning [1]. Neurocognitive functions refer to complex intellectual processes, including the ability to acquire, retain and retrieve information, as well as other cognitive abilities, such as decision making. Effective daily functioning relies on intact cognitive processing [2]. In strategic sports, such as football, players must be able to make quick and precise decisions in a dynamic and unpredictable environment [2,3,4]. Empirical evidence indicates that perceptual cognitive skills, such as visual processing, pattern recognition, pattern recall, visual search behavior, and the understanding of situational probabilities, are essential for achieving high-level performance, which can also be referred to as “game intelligence” [5,6,7]. Deficits or impairments in one or more domains of neurocognitive function may be associated with diminished athletic performance [2,8] and an elevated risk of musculoskeletal injuries [2,9].

A concussion is one factor that can impact neurocognitive function [10,11,12,13,14,15,16]. Concussion and mild traumatic brain injury are often used interchangeably [14,17]. When a concussion occurs in sports or exercise-related activities, it is usually referred to as a sport-related concussion (SRC) [13]. SRC is a traumatic brain injury that occurs following a direct or indirect impact to the head, neck or body, generating a force transmitted to the brain [13,14]. SRC does not appear on standard neuroimaging but can give rise to a range of clinical symptoms [13]. The role of concussion as a contributing factor to neurocognitive deficits and impairments has not yet been thoroughly established [18,19]. However, research has shown that SRC can trigger a neurotransmitter cascade, which may lead to neuroinflammatory responses, ultimately impairing cognitive abilities, leading to loss of emotional control, or giving rise to sleep-, somatic-, and affective symptoms [2,10,11,13,20]. Patients suffering from a SRC will generally not have loss of consciousness, but rather experience symptoms like confusion, dizziness, transient amnesia, balance problems and slow reaction time [2,10].

Professional players in contact sports such as football are exposed to head injuries. They are at risk of suffering from repetitive head impacts, which increases the risk of a complicated recovery process [2,21]. Women players have been reported to have higher concussion incidence rates than men, with typical sports involved in the incidents being football [22,23,24]. Some of the potential long-term consequences of SRC are post-concussion syndrome, dementia and chronic traumatic encephalopathy [21,25]. Women and young players, as well as players suffering from affective disorders such as depression or anxiety, are at higher risk of developing post-concussion syndrome [26]. Anxiety disorders are also more prevalent in women compared to men [27]. The combination of these risk factors, along with a higher concussion rate compared to men, makes women professional football players a particularly vulnerable group that needs attention. The first objective of the study was, therefore, to evaluate the neurocognitive functions of women professional football players. The second objective was to research potential links between previous concussion(s) and neurocognitive function. The subject is of critical importance, as it enables the optimization of performance, the prevention of injuries and long-term sequelae that could impact various aspects of a player’s life.

## 2. Materials and Methods

### 2.1. Design

An observational cross-sectional study was conducted in accordance with the Declaration of Helsinki (2024) [28]. STROBE (Strengthening the Reporting of Observational Studies in Epidemiology statement) guidelines were followed to ensure the quality of reporting. Ethical approval was provided by The Medical Ethics Review Committee of the Amsterdam University Medical Centre (Amsterdam UMC, location AMC) (Drake Football Study: NL69852.018.19|W19_171#B202169).

### 2.2. Participants

The study population consisted of women professional footballers recruited by Football Players Worldwide (FIFPRO) and their associated national unions. The following three inclusion criteria were used: (a) a professional footballer; (b) a woman; and (c) able to read and comprehend texts in English or French. In the present study, the definition for a professional footballer was that the player (i) trains to improve performances, (ii) competes in the highest or second-highest national league, and (iii) has football training and competition as the primary activity or personal interest, surpassing the time spent on other professional or leisure pursuits. No formal sample size calculation was performed, but as the preferred sample size is 50 times the number of independent variables, we strived to include at least 50 participants.

### 2.3. Neurocognitive Functions (Dependent Variables)

For the study, data were collected through “CNS vital signs” (CNS-VS), which is a computerized neurocognitive assessment tool that provides a detailed quantitative assessment of cognitive performance [29]. The program measures a broad range of cognitive functions or domains, including attention, memory, processing speed and executive function, making it a tool for detecting subtle impairments after a concussion. The program can also differentiate between healthy subjects and those suffering from various neurological or physiological diseases [1,2]. The program has been used in various sports, including football, rugby and boxing, and is available in several languages to accommodate the subjects’ native language [4,5,21,30]. The CNS-VS exhibits moderate to good reliability (test–retest intraclass correlation coefficients ranging from 0.65 to 0.88) and validity (concurrent validity correlation coefficients of up to 0.79) [29]. The program generates the patient’s scores (raw scores) and compares the patient’s scores with the standard deviation (SD) (a mean of 100 and SD of 15). Standard scores (SSs) are derived by comparing raw scores to age-matched normative data based on the general population in the United States. The standard scores were used for clinical interpretation. The domain scores are then categorized into groups: above-average domain (SS > 109), average domain (SS 90–109), low-average (SS 80–89), low (SS 70–79), and very low (SS < 70). Above-average indicates a high-functioning test subject, average indicates normal function, low-average indicates a slight deficit or impairment, below-average indicates moderate impairment, and very low testing indicates an impairment or cognitive deficit. The program includes built-in procedures to assess whether a participant is intentionally altering their test performance for secondary gain or if they simply misunderstood the testing procedures, thus ensuring the validity of the scores. Only validated scores by the program were used in the data analysis. Seven neurocognitive domains were assessed through seven CNS-VS subtests: a visual memory test, verbal memory test, Stroop test, Finger tapping test, symbol digit coding test, continuous performance test, and shifting attention test (further information provided in Appendix A).

### 2.4. Concussion (Independent Variable)

To assess the participants’ history of football-related concussions, a single question was asked: “How many concussions have you had so far during your professional football career (training and competition)?”. In this study, concussion was defined as a hit to the head (direct or transmitted) giving rise to symptoms such as cognitive or clinical symptoms, e.g., headache, nausea, vomiting, dizziness/balance problems, fatigue, trouble sleeping, drowsiness, sensitivity to light or noise, blurred vision, difficulty remembering, and difficulty concentrating. The definitions were clearly explained to the participants, and the participants were asked to consult their clinical physician or medical records to help answer the question.

### 2.5. Procedure

Information about the study was emailed to potential participants by FIFPRO and affiliated national unions, with the procedures kept confidential from the principal researcher for privacy reasons. Those interested in participating provided informed consent and were asked to complete an electronic questionnaire (CastorEDC, CIWIT B.V, Amsterdam, the Netherlands) available in both English and French. The electronic questionnaire included questions about concussion, along with several descriptive variables: age, body-weight, height, level of education, parallel activity (e.g., study, work), level of football, position on the field, number of seasons as a professional footballer, exposure to football training and matches, history of hospitalization, family history of neurological disease and self-reported global physical health (using the Patient-Reported Outcomes Measurement Information System Global Health short form; PROMIS-GH) [31]. Participants were then asked to complete the seven CNS-VS tests. To ensure privacy and confidentiality, all responses to the questionnaires and tests were coded and anonymized. Upon completion, the electronic questionnaires and test results were automatically saved on a secure server, accessible by the research team. Participants in the study volunteered and did not receive any compensation for their involvement.

### 2.6. Statistical Analysis

All data analyses were conducted using the statistical software IBM SPSS 28.0. For the primary objective, a descriptive analysis of all data from the neurocognitive function domains was conducted, with data presented as standard deviations (SDs), mean, range, frequency and percentages. Percentages of the participants suffering from a neurocognitive deficit or impairment were also calculated and categorized into four groups: 1. unlikely; 2. slightly likely; 3. moderately likely; and 4. likely to suffer from a neurocognitive deficit or impairment.

For the second objective, the number of concussions (*n* = 0–5) was presented, and a Kruskal–Wallis test was used on all neurocognitive function domains, followed by a post hoc pairwise Mann–Whitney U comparison between groups. To minimize the likelihood of false positives, a Bonferroni correction was applied to control for multiple comparisons.

## 3. Results

### 3.1. Demographics and Characteristics

A total of 74 players were recruited and answered the electronic questionnaires. The mean age of the participants was 25 years (24.95), most of them were defenders (*n* = 25), followed by forwards (*n* = 22), midfielders (*n* = 17) and goalkeepers (*n* = 10). Of the participants, 86% played in the highest national league. The physical health T-score mean was 51.5, and the mental health T-score mean was 49.6 and did not differ from the reference population. Players reported that 18 family members (24%) had been diagnosed with a neurological disease; however, none of the participants themselves reported a diagnosis of such [Table 1].

### 3.2. Neurocognitive Function

Of the 74 participants, 68 completed the CNS-VS assessment. The six participants who did not complete the CNS Vital Signs assessment were excluded from the calculations. Among them, two reported zero concussions, two reported one concussion, one reported two concussions, and one reported three concussions. The proportion of valid test results across individual neurocognitive function domains ranged between 86.5% and 91.9%. Compared to the reference population, the standard mean score (SS mean) of women professional footballers was average for 11 out of 12 neurocognitive function domains (scoring 90 or above). This suggests that overall, the average player did not show signs of neurocognitive function deficits. In the motor speed (MS) domain, the SS mean was 111.7, indicating a result above average. In the neurocognitive index (NCI) group, 94% were unlikely to have neurocognitive function deficits, whilst 3% had a slight possibility of impairment. The likelihood of impairment in each domain ranged between likely 0–4.5%, moderately likely 0–5.9%, slightly likely 3–20.7% and unlikely 66.4–94.2% [Table 2].

### 3.3. Reported Concussions

A total of 32 out of 74 participants (43%) reported having suffered from one or more concussions, with defenders being the most affected group (50%). Among defenders, 64% (*n* = 16) reported experiencing one or more concussions. This was followed by 35% of midfielders (*n* = 6), 32% of forwards (*n* = 7), and 30% of goalkeepers (*n* = 3) [Table 3]. In terms of frequency, 14 participants (19%) reported a single concussion, 12 participants (16%) reported two concussions, 3 participants (4%) reported three concussions, 1 participant (1%) reported four concussions, and 2 participants (3%) indicated having sustained five concussions [Table 3].

### 3.4. Association Between Concussion and Neurocognitive Function

Results from the Kruskal–Wallis H-test showed a statistically significant difference in the SA score across concussion history groups (*p* = 0.003). The other neurocognitive domains did not show statistical significance between groups. After conducting the Mann–Whitney U test with Bonferroni correction, only the comparisons between groups with no concussion and three concussions (*p* = 0.012), between one concussion and three concussions (*p* = 0.031), and between two and three concussions (*p* = 0.014) remained statistically significant. These findings suggest that neurocognitive dysfunction is more pronounced in participants who have suffered three concussions compared to those who reported fewer or no concussions. Results also indicate that a history of four or five concussions was not associated with a statistically significant effect on neurocognitive function, suggesting a non-linear relationship.

## 4. Discussion

This study showed that women professional footballers did not present signs of neurocognitive function deficits when compared to the general population. A significant association was observed between a history of concussion and performance in the SA domain, especially in those who have suffered from three concussions. Forty-three percent of participants reported having suffered one or more concussions, with most concussions being reported by defenders (50%).

### 4.1. Association Between Concussion and Neurocognitive Function

Participants who reported having sustained three concussions performed worse in the SA neurocognitive domain compared to those who reported none, one or two concussions. However, no significant association was observed with participants who reported four or five concussions. This non-linear finding may be due to the small sample size or to survivor bias, where individuals with greater resilience are overrepresented in the highest concussion categories. Other studies report that physical activity promotes neurogenesis. Running and regular exercise play a critical role in maintaining overall body and brain health across the lifespan, contributing to the prevention and mitigation of brain damage [32,33,34,35]. A plausible hypothesis is that footballers, by having an active lifestyle, may compensate for the potential effects of sub-concussive events. However, to determine whether a clear association exists, it is necessary to include appropriate control groups to enable more accurate comparisons.

A study conducted among male professional footballers also reported CNS-VS findings, with players having average scores on 7 out of 10 neurocognitive function domains (leaving out domains VM and VSM) [12]. This indicates lower average scores relative to the women footballers in our study, who obtained averages in 11 of the 12 neurocognitive domains. In the same study, standard scores in domains NCI, CA and PS suggested a slight to likely neurocognitive deficit or impairment. The study also reported that a history of concussion can give up to a threefold increase in the odds of having deficits in the CA or SA domains [12]. Repetitive sub-concussive events have been suggested to decrease cognitive performance, particularly memory [15,16,19]. However, this effect was not observed in our study, nor in professional male football players [12]. Instead, both in our study and in the study with male footballers, it appears that simple attention (SA) may be affected following a concussion [12]. Further research is required to validate these findings. The vulnerability observed in defenders in both studies could be clinically relevant for everyday training applications.

Another study reported that athletes who have sustained three or more sport-related concussions showed higher clinical symptom severity scores compared to athletes who have suffered fewer than three concussions [1]. Neurocognitive performance, however, did not appear to be affected. The study suggests a critical threshold of three concussions [1]. Our findings indicate a possible threshold of three concussions as well, observed in relation to neurocognitive function (SA), rather than in symptom burden, which was not measured in the present study. It is also worth noting that the neurocognitive tests used differed between the two studies.

### 4.2. Clinical Significance

Our study found that a history of three concussions was significantly associated with lower standard scores for SA. Whether this affects the everyday life of the women football players is, however, uncertain. Neurocognitive baseline data followed by follow-up measurements could provide valuable insights to prevent the progression of neurological impairment. If changes are observed over time, this could warrant a more comprehensive medical evaluation to identify the potential underlying causes, such as neurological disease, concussion, age-related decline, stress or influence of substances (e.g., drugs and/or alcohol). Such follow-up could facilitate appropriate interventions (e.g., cognitive therapy or targeted treatments), which may prohibit further disease progression. Standardizing neurocognitive testing is essential to allow reliable comparisons of results between studies or over time.

It is also worth highlighting that 6 out of 32 (19%) women footballers reported having suffered three or more concussions. If a critical threshold of three concussions can be established through repeated findings, it could be translated into practice by considering protective measures for players with a history of two concussions.

### 4.3. Research Implications

Study findings underline the need for longitudinal research on larger control cohorts to gain a deeper understanding of neurocognitive changes following a concussion. Introducing standardized individual baseline testing of neurocognitive function would increase sensitivity to post-injury changes. Monitoring variables such as recovery time, impact force, impact characteristics and symptom duration could also be of added value. Since physical activity may have near-term cognitive effects [36] and physical and mental exercise, combined with stress reduction and a healthy diet, are associated with a significant effect on cognitive function [37], tracking daily wellness scores in professional players may help account for neurocognitive function variability.

### 4.4. Study Limitations

The small sample size, especially in the higher-concussed groups, limited statistical power. Improvement of reliability in documenting concussion history (e.g., with medical records) would minimize the risk of recall bias. In addition, self-reported career concussion history may be subject to exposure misclassification, as players may forget, underreport, or fail to recognize previous concussive events. Such exposure misclassification could obscure true group differences; in particular, misreported concussion counts may dilute or distort the distribution of exposure levels, potentially contributing to the non-linear pattern observed in the Kruskal–Wallis analysis. Furthermore, other variables known to influence cognitive performance, such as sleep quality [38] and stress [39], were not accounted for in the present study. Although defenders appeared to have the highest concussion exposure, we did not include exposure proxies (e.g., seasons played and match minutes), which may confound overall head-impact exposure.

Although the CNS-VS program has been validated, the reliability and sensitivity of the neurocognitive measures should also be taken into consideration, as they may influence test results.

### 4.5. Study Strengths

Despite the limitations mentioned, the study provides valuable insights into the relationship between concussion history and neurocognitive domains. The identification of significant differences at three concussions highlights the potential clinical relevance of cumulative brain injuries. Our research may contribute to providing a guiding framework and/or serve as one of several pillars supporting future research.

## 5. Conclusions

Professional women football players showed no significant impairments in overall neurocognitive functioning. However, a history of three concussions was significantly associated with lower standardized scores within the simple attention domain. Future research and long-term follow-up are needed to better identify and understand patterns over time.

## Figures and Tables

**Table 1 sports-13-00448-t001:** Demographics and characteristics of participants. Number (N), percentage (%), standard deviation (SD).

Demographics	Age (Mean and SD)	25	3
Football characteristics	Seasons played (Mean and SD)	6	3
	Playing position (*n* and %)		
	Goalkeeper	10	14
	Defender	25	34
	Midfielder	17	23
	Forward	22	30
	Career level (*n* and %)		
	Highest national level	62	84
	Second-highest national level	6	8
	Other levels	6	8
PROMIS-GH	Physical Health T-Score	52	5
T-scores (Mean and SD)	Mental Health T-score	50	7
Neurological Disease (N and %)	Diagnosed player	0	0
	Diagnosed family member	18	24
	Dementia	4	5
	Parkinson’s	3	4
	Alzheimer’s	6	8
	Epilepsy	2	3
	Dementia + Parkinson’s	1	1
	Alzheimer’s + Parkinson’s	2	3

**Table 2 sports-13-00448-t002:** Neurocognitive domain, validity, deficits, and means. Standard score mean (SS mean), standard deviation (SD), per centage (%), neurocognitive index (NCI), composite memory (CM), verbal memory (VM), reaction time (RT), complex attention (CA), cognitive flexibility (CF), processing speed (PS), executive functioning (EF), simple attention (SA), motor speed (MS), visual memory (VSM), psychomotor speed (PMS).

Domain	Valid %	Valid *n*	Unlikely %	Slight %	Moderate %	Likely %	SS Mean	SD
NCI	87	64	94	3	0	0	99	18
CM	91	67	82	15	2	0	100	17
VM	91	67	83	13	3	0	102	17
RT	89	66	66	16	3	0	98	16
CA	88	65	79	12	3	4	98	18
CF	89	66	80	7	6	5	99	18
PS	91	67	74	21	2	3	97	17
EF	92	68	82	9	6	3	101	14
SA	91	67	82	10	3	5	96	26
MS	92	68	94	5	2	0	112	15
VSM	91	67	82	9	6	2	98	18
PMS	91	67	96	5	0	0	108	14

**Table 3 sports-13-00448-t003:** Number of concussions (total and per position).

	Total	0	1	2	3	4	5
Total (*n* = 74)	32	42	14	12	3	1	2
Goalkeeper (*n*= 10)	3	7	1	1	1	0	0
Defender (*n*= 25)	16	9	6	8	1	0	1
Midfielder (*n*= 17)	6	11	5	0	0	0	1
Forward (*n*= 22)	7	15	2	3	1	1	0

## Data Availability

All data analyses of data collected are presented in the manuscript. Raw data can be requested from the corresponding author and will be shared upon reasonable request.

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
