# Peer review of "The Relationship Between Neurocognitive Function and Concussion in Women Professional Football Players: A Cross-Sectional Study"

_sports, 2025, doi:10.3390/sports13120448_

Round 1
Reviewer 1 Report
Comments and Suggestions for Authors
please see attached

Author Response
Reviewer 1:
Reviewer’s comment: First, I would like to recognize the authors for the work they have presented. There is indeed a need for more studies on professional female soccer players who are underrepresented, especially in concussion research (despite the high concussion incidence rates we see in female players). Furthermore, I would like to recognize the authors for using the CNS vital signs test, which is a standardized test for neurocognitive assessment with documented reliability and validity. Last but not least, the study is clearly written, with an excellent flow and support. The finding that players with a history of three concussions showed significant deficits in the simple attention domain can be of great importance. The abstract contains all the necessary information, and it is very well-presented. The introduction contains all the relevant scientific literature and leads to the objective of the study. Terms such as neurocognition, neurocognitive function and concussions are clearly explained and defined. The methodology section is well-presented. It is presented in detail so that someone can replicate the study, and it is scientifically supported.
Authors reply: We thank the reviewer for your kind words and time reviewing our manuscript, as well as for the comments and suggestions. We tried to alter the manuscript appropriately, considering the comments and suggestions. We sincerely hope that the reviewer is satisfied with the changes we have made.
Reviewer’s comment: Tables. I would present the result using a full stop rather than a comma.
Authors reply: Thank you for your input. We have now altered the tables accordingly.
Reviewer’s comment: Table 4 needs improvement. While I do understand that you are presenting the number of concussions per position, I am not sure what you are presenting there.
Authors reply: Thank you for suggestion. We have combined Table 3 and Table 4 in order to make clearer what we want to present, namely the number of concussions per position.
Reviewer’s comment: I find that most of the cross-sectional studies have limited causal inference, as it is impossible to determine whether the observed cognitive differences are actually the result of concussion or reflect pre-existing differences. However, this limitation does not influence the significance of the study. Furthermore, the fact that only six players had equal to, or more than three concussions is another limitation. In addition, the retrospective self-report of concussions is subject to recall bias, as some players may forget mild concussions or misidentify non-concussive symptoms. I also consider the absence of data on factors such as sleep quality, fatigue, menstrual cycle phase, medication use and stress to be a limitation, as these variables can significantly influence cognitive test performance. These physiological and psychological factors are known to have measurable effects on cognitive performance. For example, inadequate sleep and fatigue can impair attention, processing speed, and executive function, while hormonal fluctuations across the menstrual cycle may influence mood and cognitive stability. Similarly, medication use or elevated stress levels can alter reaction time, concentrations and memory performance. Without accounting for these variables, it becomes difficult to determine whether observed cognitive differences are truly attributable to concussion history or to temporary, non-injury-related influences. I would suggest adding some of these factors to the limitations section. Having said that, the study still makes a valuable contribution to the limited literature on female soccer players’ concussion-related neurocognitive outcomes. However, its crosssectional design, small sample, self-reported data and lack of controls limit the strength of the conclusions. Future studies should focus on a prospective and longitudinal approach to better capture the effects of concussion on cognitive health in female soccer players.
Authors reply: We appreciate the reviewer’s valuable point and have added the following to the limitations section in the manuscript: “Furthermore, other variables known to influence cognitive performance, such as sleep quality [35] and stress [36], were not accounted for in the present study”
Reviewer 2 Report
Comments and Suggestions for Authors
This cross-sectional survey of professional women’s footballers uses CNS Vital Signs (CNS-VS) to profile neurocognition and examines associations with self-reported sport-related concussion history. Sixty-eight players completed testing; group means were largely in the age-normative range, with motor speed elevated. The main between-group effect was poorer Simple Attention in the subset reporting three prior concussions (post-hoc contrasts remained significant after Bonferroni correction). The paper is timely and clinically relevant, and the player-facing recruitment is a strength. That said, several methodological and statistical choices limit interpretability. Below I outline specific, actionable revisions—most are feasible within the current dataset.
The primary independent variable is a single self-report item (“How many concussions…?”), with broad symptom wording and recall to the entire professional career. This sets the stage for recall bias and exposure misclassification, and it prevents any inference about temporal ordering. Please justify using a single item rather than a structured instrument or medical record confirmation, and discuss how misclassification would bias your Kruskal-Wallis results (likely toward null, but differential misclassification could also distort non-linear patterns).
Participants completed CNS-VS in English or French; your norms are U.S. age-based. Please state explicitly whether language version effects were tested (English vs. French), whether the interface language matched each player’s dominant language, and whether any country/education adjustments are appropriate. Also, you report “valid test results” per domain of ~86.5–91.9%; what were your a priori criteria for validity flags and how were invalid domain scores handled (listwise vs. pairwise deletion)? A sensitivity analysis re-including flagged tests (or excluding all flagged domains) would show robustness.
The statement that the “preferred sample size was 50 times the number of the independent variable” is unconventional for group-comparison nonparametrics and does not constitute a power analysis. Please replace with an a priori power calculation for your primary outcome (e.g., detectable standardized difference in Simple Attention across concussion bins with α adjusted for multiplicity), or reframe as exploratory with appropriate caution.
You report more concussions among defenders and highlight their potential vulnerability. Could you add a position-adjusted analysis of Simple Attention vs. concussion count? Also consider exposure proxies (seasons played, match minutes if available) to reduce confounding by overall head-impact exposure.
The paper occasionally implies a threshold at “three concussions.” With a cross-sectional design, uneven bin sizes, and multiple comparisons, please keep language to associations and “hypothesis-generating.” Consider adding raincloud/violin plots for Simple Attention by concussion count, with raw data points, medians, and 95% CIs, and include effect sizes for key pairwise contrasts.
Why were neuroimaging or electrophysiological studies on elite athletes’ cognitive function not considered in framing your hypotheses or interpreting your null behavioral findings? The discussion could be strengthened by referencing and contrasting findings from elite athlete brain studies that combine behavioral and neural measures. Several landmark studies have shown functional and structural neuroplasticity linked to superior stimulus-driven visual attention, cognitive control, inhibition, and working memory in Olympic elite athletes.
Figures/tables could benefit from clearer visualization of confidence intervals or distribution spread rather than only means and SDs.
Author Response
Reviewer 2:
Reviewer’s comment: This cross-sectional survey of professional women’s footballers uses CNS Vital Signs (CNS-VS) to profile neurocognition and examines associations with self-reported sport-related concussion history. Sixty-eight players completed testing; group means were largely in the age-normative range, with motor speed elevated. The main between-group effect was poorer Simple Attention in the subset reporting three prior concussions (post-hoc contrasts remained significant after Bonferroni correction). The paper is timely and clinically relevant, and the player-facing recruitment is a strength. That said, several methodological and statistical choices limit interpretability. Below I outline specific, actionable revisions—most are feasible within the current dataset.
Authors reply: We thank the reviewer for your time reviewing our manuscript, as well as for the comments and suggestions. We tried to alter the manuscript appropriately, considering the comments and suggestions. We sincerely hope that the reviewer is satisfied with the changes we have made.
Reviewer’s comment: The primary independent variable is a single self-report item (“How many concussions…?”), with broad symptom wording and recall to the entire professional career. This sets the stage for recall bias and exposure misclassification, and it prevents any inference about temporal ordering. Please justify using a single item rather than a structured instrument or medical record confirmation, and discuss how misclassification would bias your Kruskal-Wallis results (likely toward null, but differential misclassification could also distort non-linear patterns).
Authors reply: Thank you for this insightful comment. We have now incorporated the following into the limitations section of the study: “In addition, self-reported career concussion history may be subject to exposure misclassification, as players may forget, underreport, or fail to recognize previous concussive events. Such exposure misclassification could obscure true group differences; in particular, misreported concussion counts may dilute or distort the distribution of exposure levels, potentially contributing to the non-linear pattern observed in the Kruskal–Wallis analysis.”
Reviewer’s comment: Participants completed CNS-VS in English or French; your norms are U.S. age-based. Please state explicitly whether language version effects were tested (English vs. French), whether the interface language matched each player’s dominant language, and whether any country/education adjustments are appropriate. Also, you report “valid test results” per domain of ~86.5–91.9%; what were your a priori criteria for validity flags and how were invalid domain scores handled (listwise vs. pairwise deletion)? A sensitivity analysis re-including flagged tests (or excluding all flagged domains) would show robustness.
Authors reply: Thank you for your comment and suggestions. We have added some information related to language in our manuscript. CNS Vital Signs includes embedded validity indicators that automatically flag potentially invalid test or domain scores (e.g., due to poor effort or misunderstanding). As stated in the CNSVC manual and available on the CNS Vital Signs website, when a domain fails its validity indicator the recommended action is to retest that specific domain, unless the clinician judges the score to reflect true impairment. CNSVS does not prescribe a statistical rule such as listwise or pairwise deletion; rather, decisions about how to handle invalid scores are left to the researcher.
Reviewer’s comment: The statement that the “preferred sample size was 50 times the number of the independent variable” is unconventional for group-comparison nonparametrics and does not constitute a power analysis. Please replace with an a priori power calculation for your primary outcome (e.g., detectable standardized difference in Simple Attention across concussion bins with α adjusted for multiplicity), or reframe as exploratory with appropriate caution.
Authors reply: According to the reviewer’s comment, we altered our manuscript.
Reviewer’s comment: You report more concussions among defenders and highlight their potential vulnerability. Could you add a position-adjusted analysis of Simple Attention vs. concussion count? Also consider exposure proxies (seasons played, match minutes if available) to reduce confounding by overall head-impact exposure.
Authors reply: Thank you for this remark. We agree with the reviewer that we need to mention exposure properties, which is why the following is added to the limitations section in the present study: “Although defenders appeared to have the highest concussion exposure, we did not include exposure proxies (e.g., seasons played, match minutes), which may confound overall head-impact exposure.”
Reviewer’s comment: The paper occasionally implies a threshold at “three concussions.” With a cross-sectional design, uneven bin sizes, and multiple comparisons, please keep language to associations and “hypothesis-generating.” Consider adding raincloud/violin plots for Simple Attention by concussion count, with raw data points, medians, and 95% CIs, and include effect sizes for key pairwise contrasts.
Authors reply: Thank you for this comment. We have revised the text to ensure the findings are framed strictly as associations and clearly described as hypothesis-generating, with no implied threshold. We also note that future longitudinal work is needed to explore this further. Regarding the suggested raincloud/violin plots, we appreciate the recommendation but have chosen not to add additional figures in order to keep the focus on the primary analyses.
Reviewer’s comment: Why were neuroimaging or electrophysiological studies on elite athletes’ cognitive function not considered in framing your hypotheses or interpreting your null behavioral findings? The discussion could be strengthened by referencing and contrasting findings from elite athlete brain studies that combine behavioral and neural measures. Several landmark studies have shown functional and structural neuroplasticity linked to superior stimulus-driven visual attention, cognitive control, inhibition, and working memory in Olympic elite athletes.
Authors reply: Thank you for your suggestion. We agree that neuroimaging and electrophysiological studies provide important insights into neural adaptations in elite athletes and could meaningfully strengthen the interpretation of behavioral findings. While these methods would indeed offer added value when available and feasible, they were beyond the scope of the current study and not accessible within our research setting.
Reviewer’s comment: Figures/tables could benefit from clearer visualization of confidence intervals or distribution spread rather than only means and SDs.
Authors reply: Thank you for this helpful suggestion. We agree that visualizing confidence intervals can be informative; however, we have chosen to retain the current figure format to maintain consistency with the primary analytic approach and to keep the presentation focused on the main results.
Reviewer 3 Report
Comments and Suggestions for Authors
Thank you for the opportunity to review your manuscript, “The relationship between neurocognitive function and concussion in women professional football players: a cross-sectional study.”
The study aims was to evaluate the neurocognitive functions of women professional football players and research potential links between previous concussion(s) and neurocognitive function.
This is a cross-sectional observational study aimed at determining the neurocognitive functions of professional female soccer players and exploring their possible connection with concussions.
Cognitive function was assessed in 68 participants using the "CNS Vital Sign" (a computerized neurocognitive assessment tool). The results show that the players exhibit scores in the average range across 11 of the 12 neurocognitive domains, indicating that they do not present significant neurocognitive deficits compared to the reference population.
Forty-three percent (n=32) of the participants had suffered one or more concussions, with the defenders being the most affected group, reaching 50%. Although a significant neurocognitive deficit was not observed overall, the players with a history of three or more concussions showed significant deficits in the simple attention domain.
This article shares the ethics committee number with another study, "Unmasking mental health symptoms in female professional football players: a 12-month follow-up study," which also shares some of the same authors. Both articles utilize the same sample of professional female football (soccer) players, share some of the authors, and the data collection for both was carried out using electronic questionnaires.
Lines 43-46: While it is acceptable to include empirical evidence found in a book, we may be more interested in knowing what the scientific evidence (from peer-reviewed literature) states regarding this topic.
Line 50: Is this acronym (mTBI) necessary if it only appears once in the entire text?
Line 68: Is this acronym (PCS) necessary if it only appears twice in the text?
Line 123: The authors state that more information is available in an appendix, but I cannot find this information.
Tables: Standard formatting requires the title to be placed at the top of the table, and the legend for abbreviations should be placed at the bottom of the table.
Author Response
Reviewer 3
Reviewer’s comment: Thank you for the opportunity to review your manuscript, “The relationship between neurocognitive function and concussion in women professional football players: a cross-sectional study.”
The study aims was to evaluate the neurocognitive functions of women professional football players and research potential links between previous concussion(s) and neurocognitive function.
This is a cross-sectional observational study aimed at determining the neurocognitive functions of professional female soccer players and exploring their possible connection with concussions.
Cognitive function was assessed in 68 participants using the "CNS Vital Sign" (a computerized neurocognitive assessment tool). The results show that the players exhibit scores in the average range across 11 of the 12 neurocognitive domains, indicating that they do not present significant neurocognitive deficits compared to the reference population.
Forty-three percent (n=32) of the participants had suffered one or more concussions, with the defenders being the most affected group, reaching 50%. Although a significant neurocognitive deficit was not observed overall, the players with a history of three or more concussions showed significant deficits in the simple attention domain.
This article shares the ethics committee number with another study, "Unmasking mental health symptoms in female professional football players: a 12-month follow-up study," which also shares some of the same authors. Both articles utilize the same sample of professional female football (soccer) players, share some of the authors, and the data collection for both was carried out using electronic questionnaires.
Authors reply: We thank the reviewer for your time reviewing our manuscript, as well as for the comments and suggestions. We tried to alter the manuscript appropriately, considering the comments and suggestions. We sincerely hope that the reviewer is satisfied with the changes we have made
Reviewer’s comment: Lines 43-46: While it is acceptable to include empirical evidence found in a book, we may be more interested in knowing what the scientific evidence (from peer-reviewed literature) states regarding this topic.
Authors reply: Thank you for your comment. Additional scientific references have been included to support this statement.
Reviewer’s comment: Line 50: Is this acronym (mTBI) necessary if it only appears once in the entire text?
Authors reply: Thank you for the observation; we have now addressed this in the manuscript.
Reviewer’s comment: Line 68: Is this acronym (PCS) necessary if it only appears twice in the text?
Authors reply: We thank the reviewer for the comment and have revised the text accordingly.
Reviewer’s comment: Line 123: The authors state that more information is available in an appendix, but I cannot find this information.
Authors reply: Thank you for the comment. The information is located in Appendix X, and we have revised the manuscript to make this more clearly visible.
Reviewer’s comment: Tables: Standard formatting requires the title to be placed at the top of the table, and the legend for abbreviations should be placed at the bottom of the table.
Authors reply: We appreciate the reviewer’s observation. The headings and table descriptions were prepared in accordance with MDPI requirements; nonetheless, we have reviewed the formatting again to ensure that everything is clearly presented.
Round 2
Reviewer 2 Report
Comments and Suggestions for Authors
Authors should review research on neurocognitive functions in elite athletes and cite relevant studies, especially given the substantial literature on neurocognitive function and concussion in female open-skill athletes.
Author Response
Reviewer’s comment: Authors should review research on neurocognitive functions in elite athletes and cite relevant studies, especially given the substantial literature on neurocognitive function and concussion in female open-skill athletes.
Authors reply: Thank you for your comment. Additional scientific references have been included to support this statement.